# A Fast and Efficient Procedure of Iron Species Determination Based on HPLC with a Short Column and Detection in High Resolution ICP OES

**DOI:** 10.3390/molecules28114539

**Published:** 2023-06-03

**Authors:** Aleksandra Orłowska, Jędrzej Proch, Przemysław Niedzielski

**Affiliations:** 1Department of Analytical Chemistry, Faculty of Chemistry, Adam Mickiewicz University, Uniwersytetu Poznańskiego 8, 61-614 Poznań, Poland; aleksandra.orlowska@amu.edu.pl (A.O.); przemyslaw.niedzielski@amu.edu.pl (P.N.); 2Faculty of Archaeology, Adam Mickiewicz University, Uniwersytetu Poznańskiego 7, 61-614 Poznań, Poland; 3Interdisciplinary Research Group Archaeometry, Faculty of Archaeology and Faculty of Chemistry, Adam Mickiewicz University, Uniwersytetu Poznańskiego 7–8, 61-614 Poznań, Poland

**Keywords:** hyphenated technique, HPLC–ICP hrOES, short column, iron speciation, archaeometry, geology

## Abstract

The optimization and application of a new hyphenated procedure for iron ionic speciation, i.e., high performance liquid chromatography (HPLC) with short cation–exchange column (50 mm × 4 mm) coupled to high resolution inductively coupled plasma optical emission spectrometry (ICP hrOES), is presented in this paper. Fe(III) and Fe(II) species were separated on the column with the mobile phase containing pyridine–2,6–dicarboxylic acid (PDCA). The total time of the analysis was approx. 5 min, with a significantly low eluent flow rate (0.5 mL min^−1^) compared to the literature. Additionally, a long cation-exchange column (250 mm × 4.0 mm) was used as reference. Depending on the total iron content in the sample, two plasma views were chosen, e.g., an attenuated axial (<2 g kg^−1^) and an attenuated radial. The standard addition method was performed for the method’s accuracy studies, and the applicability was presented on three types of samples: sediments, soils, and archaeological pottery. This study introduces a fast, efficient, and green method for leachable iron speciation in both geological and pottery samples.

## 1. Introduction

Iron is an element commonly found in various types of soils, and occurring mainly in two oxidation states: Fe(II) and Fe(III) [1]. Both of these species have significant but different roles in the biochemical processes in the environment, and it is therefore important to distinguish them. Redox reactions of this element take part in geochemical transformations and photosynthesis. Moreover, Fe oxides are heavy metal sorbents in the soil, and the element is a co-factor of enzymes in plants [2,3,4]. Iron is also an essential element for the human body, as the right content prevents, to take just one example, anaemic diseases. Due to this, food fortification approaches exist that allow for the supplementation of iron, especially in developing countries with high iron deficiency among the population. The aim of such a program is to obtain modified crops that can efficiently accumulate this element from the soil [5], and due to the different uptake mechanisms, i.e., in the form of Fe(II) or Fe(III) complexes [6], knowledge about iron speciation in soils is required.

Pottery is another example of a material that is being studied for iron species, e.g., in black glaze of Chinese pottery [7], Sicilian “proto–majolica” pottery [8], Sicilian antique pots [9], and Brazilian [10] and Polish ceramics [11]. The oxidation state of the element depends on the used clay and the method of firing (temperature, technique). Although both iron species are usually present in the material or coating, a higher content of Fe(III) is observed in red–orange ceramics and Fe(II) in dark grey vessels. Therefore, the Fe(III)/Fe(II) ratio is valuable and meaningful, as it may indicate the method of ceramic production [11].

In the literature, there are numerous methods for iron speciation. Colorimetric analysis is simple and fast, and it is conducted with more accessible instruments (e.g., a portable photometer). Specific reagents are required for this, which, as a result of the reaction, change the colour of the solution with an intensity adequate to the iron content in the sample. Fortune and Melon [12] presented a method for Fe(II) determination with 1,10–phenantroline, which has been used, to take just one example, in the analysis of Sicilian pottery [9]. Ferrozine assay, described by Stookey [13], is another colorimetric analysis for Fe(II), performed in marine waters [14], sediments, and glacial samples [15,16,17]. 2,2′–bipirydyl and potassium thiocyanate are used for Fe(II) and Fe(III) determination, respectively, and they were applied in food samples [18], soil [19], sediments [20], and ceramics [11]. Unfortunately, colorimetric methods are not very sensitive, as the absorbance may potentially be significantly falsified due to the presence of other compounds, e.g., Fe(III) in Fe(II) determination [21]. In addition, 1,10–phenanthroline and 2,2′–bipirydyl are toxic, which, in a large series of samples, increases the health risk of the analyst [22]. For determination of ionic iron species, Fe(II) and Fe(III), ion exchange chromatography was often chosen by many authors who used the Dionex IonPac CS5A column (250 mm × 4.0 mm, i.d.), combined with short column Dionex IonPac CG5A (50 mm × 4.0 mm i.d.) (both Thermo Fisher Scientific, Waltham, MA, USA) [23,24,25,26,27]. The most important compound for iron speciation analysis, based on these columns, is pyridine-2,6-dicarboxylic acid (PDCA), which is a complexing agent. Other reagents are variable and depend on chosen methods as well as analysed materials, e.g., lithium hydroxide, sodium sulfite, sodium chloride [28], acetic acid, sodium acetate, ascorbic acid [24], potassium chloride, formic acid, potassium hydroxide [27] or potassium sulfate, potassium hydroxide, and, finally, formic acid, which is a commercially available mobile phase dedicated to the CS5A column [23,25,29,30,31,32]. Commercial eluent is often used in the literature, usually with the post-column reagent for HPLC with UV–Vis detection. The PDCA eluent with 4–(2–pyridylazo) resorcinol (PAR) was frequently used for iron speciation analysis [23,24,25,28,29,30,31,32], too. Luminol was another example of the post-column reagent used for the chemiluminescence detection of both Fe(II) and Fe(III) [27,33]. Nevertheless, the presence of the post-column reagent increases the amount of waste, which means a significant volume in the case of a large series of samples. Therefore, other systems that have a direct connection between a column and a detector are required. This gap can be filled by hyphenated techniques, e.g., high performance liquid chromatography with inductively coupled plasma optical emission spectrometry (HPLC–ICP OES), or with inductively coupled plasma mass spectrometry (HPLC–ICP MS). In the area of these systems used for iron speciation analysis, two studies were based on ICP OES detector, preceded by a short column (CG5A) [34] or a long column (CS5A) [35]. In the case of the ICP MS detector, only two studies were found that used the CG5A column and the CS5A column with ammonium citrate and PDCA as a mobile phase [26] and also with commercial PDCA-based eluent [36]. However, mass spectrometry is typically used for samples, where iron content is much lower in comparison to soils or ceramics. Fe(II) and Fe(III) were determined in clinical samples by capillary electrophoresis (CE) with ICP MS [37], and they were determined in beverages with the use of a short anion-exchange column [38]. Other techniques of separation have also been used with ICP MS as a detector, e.g., size exclusion chromatography (SEC), although these techniques are useful in protein determination [39].

In this study, the first combination of a short cation–exchange HPLC column (50 mm × 4 mm) with a detection in high resolution ICP OES (ICP hrOES) was presented. The novel system was properly optimized and then compared with the previously presented HPLC–ICP OES [35]. The applicability of the method was performed with three types of real samples: sediments, soils, and archaeological pottery. The achieved parameters, i.e., limit of detection, accuracy, precision, and standard addition recovery, were critically evaluated. The total content of iron obtained by ICP hrOES was controlled with inductively coupled plasma mass spectrometry with Integrated Collision–Reaction Cell (ICP (iCRC)MS). The introduction of short column HPLC coupled to ICP hrOES was reported as a missing tool in hyphenated techniques for the determination of the inorganic iron species, Fe(II) and Fe(III).

## 2. Experimental

### 2.1. Gases and Reagents

High-pure argon (N–5.0, purity 99.999%) was purchased from Linde Gaz Polska (Kraków, Poland). High-pure deionized water (≥18 MΩ cm resistivity), which was obtained from a Milli–Q water purification system (Merck Millipore, Darmstadt, Germany), was used to prepare all of the solutions. Ferrous ammonium sulfate hexahydrate and ferric ammonium sulfate dodecahydrate were obtained from Acros Organics (Thermo Fisher Scientific, Waltham, MA, USA). Stock standard solutions (1000 mg L^−1^ in a single solution) were prepared by dissolving appropriate amounts of the powder in water. Less concentrated standard solutions obtained by a dilution of the stock solutions were prepared daily (due to the occurrence of disproportions). Stock standard solutions were stored in polypropylene bottles (Nalgene, Rochester, NY, USA) at 4 °C in darkness, and they were periodically controlled during the analysis of total iron content.

Commercial mobile phase PDCA eluent, MetPac concentrate (Thermo Fisher Scientific) was tested and compared with the modified version. The original product contains pyridine–2,6–dicarboxylic acid (PDCA), KOH, K_2_SO_4_, and HCOOH in concentrations of 7.0, 66, 5.6, and 74 mmol L^−1^, respectively, with a pH of 4.2 ± 0.2. The modified mobile phase was prepared by mixing appropriate volumes of PDCA and HCOOH and then dissolving appropriate amounts of NaOH and Na_2_SO_4_ (all of the reagents were from Merck) in order to obtain the same concentrations and the same pH as in the commercial eluent. A total of 100 mmol L^−1^ sodium sulfite was used as a column-cleaning solution, and prepared by dissolving Na_2_SO_3_ (Merck) in water. As a extractant, 2 mol L^−1^ HCl was used, obtained from 37% HCl (Merck).

### 2.2. Instrumentation

The HPLC system was constructed from a HPLC pump, Varian Prostar 210 (Agilent Technologies, Santa Clara, CA, USA), and a cation–exchange HPLC short column, Dionex IonPac CG5A, 50 mm × 4.0 mm i.d. (Thermo Fisher Scientific). The injection volume was 200 µL, and the eluent flow rate (0.5 mL min^−1^) was isocratic. Eluent was transferred directly from the column to the nebulizer by the use of PEEK and capillary tubing. The cation exchange HPLC column, Dionex IonPac CS5A, 250 mm × 4.0 mm i.d. (Thermo Fisher Scientific) was used as a reference column. Each column was prophylactically conditioned with a cleaning solution (100 mmol L^−1^ sodium sulfite) for 30 min each day prior to the analysis.

The sequential ICP hrOES, Plasma Quant 9000 (AnalytikJena, Jena, Germany) was used as a detector of the hyphenated system. The Triple Peltier system was used for CCD detector cooling to −10 °C. Radio frequency (RF) power was set up at 1200 W. Plasma, nebulizer (OneNeb, Agilent) and auxiliary argon flow rates were 12, 0.7, and 1.0 L min^−1^, respectively. Three plasma views were investigated (attenuated axial, radial, and attenuated radial). A peristaltic pump of the spectrometer was used only to drain the waste. Emission lines were chosen based on both the literature and the instrument software data. Full experimental parameters are shown in Table 1. Additionally, inductively coupled plasma mass spectrometry with Integrated Collision–Reaction Cell (ICP (iCRC)MS), with helium as a collision gas, was used as a reference technique determining total iron content.

### 2.3. Sample Preparation

The extraction procedure with 2 mol L^−1^ hydrochloric acid solution (HCl) was carried out in accordance with a method described previously [18]. Firstly, soils and sediments were dried at room temperature and potshards were milled using a mortar grinder, RM 200 (Retsch, Haan, Germany). Afterwards, 1.00 ± 0.01 g of sample was put in a conical flask, and 20 mL of 2 mol L^−1^ hydrochloric acid solution was added. When the flask was set with a reflux condenser, it was heated up to 80 °C and kept for 30 min. Next, 20 mL of deionized water was added and held at 80 °C for another 30 min. After this, the flask was taken from a hot plate for cooling. The sample was then filtered quantitatively to the plastic test tube through a polyethylene funnel with filter paper (which had been already rinsed with 200 mL of deionized water) and filled up with water to the volume of 50 mL.

## 3. Results and Discussion

### 3.1. Optimization

Although the conditions and the parameters of this technique (HPLC–ICP hrOES) were based on the previously optimised system (HPLC–ICP OES) [35], it was constructed from this basis, although all of the crucial parts (i.e., a HPLC pump, a HPLC column, and a detector) were different. Therefore, significant improvements have been proposed both in terms of the chromatographic run and the spectrometric detection. The first ones concerned the use of the short column (Dionex IonPac CG5A) instead of the long column (Dionex IonPac CS5A), which was used only as a reference. In the case of detection, a high resolution optical emission spectrometer was used, which allowed for an intensity measurement of selected wavelength with a spectral resolution lower than 2 pm per pixel at 200 nm. The selectivity was possible due to the presence of the double monochromator on the basis of a prism and also an echelle grating monochromator. According to the above, high resolution ICP OES was a much more resistant detector to signal interferences than this conventional one. All of the parameters are presented in Table 1.

#### 3.1.1. Chromatographic Run

The PDCA eluent, which consists of 74 mmol L^−1^ HCOOH, 66 mmol L^−1^ KOH, 7.0 mmol L^−1^ PDCA, 5.6 mmol L^−1^ K_2_SO_4_ and is adjusted to a pH of 4.2 ± 0.2, was tested by various authors [23,29,30,40] and then commercialized. On the one hand, a higher concertation of neutral salt can shorten retention time (RT) as well as decrease the mutual RT ratio of both Fe(II) and Fe(III) [33]. On the other hand, the increase of the salt load (e.g., chlorides or sulfates) can be detrimental to the sample introduction system and the plasma stability of the ICP spectrometers [26]. During the research, it was also noticed that the modified PDCA eluent, containing NaOH and Na_2_SO_4_, ensured more stable work of the HPLC system in comparison to the commercial PDCA eluent (containing KOH and K_2_SO_4_). According to this, a degasser should be an indispensable accessory when using the commercial PDCA eluent. However, the high content of sodium compounds in the mobile phase necessitates regular cleaning of both the ICP torch and the cone, otherwise it may damage the burner. Nevertheless, this modification of the mobile phase was recognized as a significant improvement of the commercially available eluent, and it was used for further research.

A cation–exchange long column (CS5A), combined with short column (CG5A), has been chosen by many authors [23,26,28,31,33]. Surprisingly, there is not much research in which CG5A has been used as a main column for iron species separation. Only Fernsebner et al. applied CG5A, but biological samples were analysed with the eluent containing 1.5 mmol L^−1^ PDCA, 10 mmol L^−1^ Tris–HAc and 500 mmol L^−1^ NH_4_Ac [34]. Three other studies were found where a short column was used for iron speciation analysis, but different types of column were used in these studies, i.e., reversed phase (Rad–Pak, reversed–phase [41] and anion exchange (Protein–Pak DEAE–5PW [42], Metrosep A Supp 10 [38]). The combination of two CG5A columns was tested for iron speciation in cerebrospinal fluid, but it resulted in weak Fe(III) retention and Fe(II) partial sticking [26]. Due to the presence of two different opinions for CG5A efficiency regarding the iron speciation method, a comparison of CS5A and CG5A on a real sample was performed in this study (Figure 1).

Firstly, the conditions were controlled using the same column (CS5A) and eluent flow rate (2 mL min^−1^), as was done before [35]. As expected, iron species separated well, resulting in 480 s of total analysis time, and retention times (RT) were 157 ± 3 s and 448 ± 10 s for Fe(III) and Fe(II), respectively. In the case of the short column (CG5A), the adequate separation of peak areas was also achieved, although RTs were significantly shorter, i.e., 123 ± 4 s for Fe(III) and 203 ± 7 s for Fe(II). Moreover, the total time of analysis was 240 s at an eluent flow rate 0.5 mL min^−1^. This means that the analysis time was halved with a short column and the consumption of the mobile phase was significantly lower. These features are beneficial for both the instrument and the environment.

In the literature, the total time of analysis was slightly longer than that obtained in this study, but Fe(III) was usually eluted faster (approx. 60 s). Using CG5A and a different PDCA-based eluent with a flow rate at 0.8 mL min^−1^, Fernsebner et al. obtained 300 s of total analysis, while RTs were 60 s and 250 s for Fe(II) and for Fe(III), respectively [34]. Using an anion–exchange short column with a mobile phase flow rate at 0.8 mL min^−1^, Wolle et al. achieved even shorter RTs, 60 s and 130 s for Fe(III) and Fe(II), respectively, although the total time of the analysis was similar to this study (250 s) [38]. This difference might be related to the different mechanism of species separation, eluent flow rate, and tubing lengths. Summarizing the comparison of CS5A and CG5A columns, further research was only conducted with CG5A column.

#### 3.1.2. Spectrometric Detection

During the optimization of the chromatographic run, the influence of the standard solution matrix on Fe(III) and Fe(II) intensity was observed, and it was therefore tested. Aqueous solutions were compared to standards containing 0.01 mol L^−1^ HCl (which was equal to the acid concentration in a sample extract after its dilution). Fe(II) peaks were almost identical in both solutions, but in the case of Fe(III), the difference was significant, giving a much higher, narrower, and compact signal in 0.01 mol L^−1^ HCl (Figure 2). This influence of the matrix in the preparation of the calibration standard was probably observed for the first time in the literature. According to this, less concentrated iron standards were prepared in 0.01 mol L^−1^ HCl as the imitation of a diluted sample extract.

Other parameters, such as analytical wavelength, Fe II 238.204 nm (ionic), Fe II 261.187 nm (ionic) and Fe II 262.167 nm (ionic), and plasma view (attenuated axial, radial and attenuated radial), were also investigated. It should be noted that three modes of plasma view were optimized for the first time in iron speciation analysis based on hyphenated techniques. Due to the high iron content in the samples, the aim was to obtain peaks within the range of measured intensities without losing any of the two iron species at dilution factors (DFs), ranging from 50 to 100.

Firstly, an axial attenuated plasma view was investigated at the most sensitive emission line, Fe II 238.204 nm. Under these conditions, it was only possible to measure samples containing less than 2 g kg^−1^ of total iron with 50- or 100-fold dilution. However, a sample matrix such as this with sediment, soil, and pottery usually contains higher amounts of total iron. In this case, higher dilution factors are desired (ranging from 150 to 300), although it leads to both higher detection limits and measurement uncertainties, which are not preferred. Therefore, other less sensitive emission lines were taken into the consideration according to the spectrometer’s software (ASpectPQ 1.3.0). In the case of Fe II 262.167 nm and the attenuated axial view, peak heights were minimal at 50- or 100-fold dilution. Presumably, it would even be possible to analyse extracts directly under these conditions. Nevertheless, a fixed dilution is desirable to extend the column lifetime (slightly unifying the pH of the extracts and the eluent). Another recommended wavelength, Fe II 261.187 nm, was also tested, but the results were similar to Fe II 238.204 nm.

In order to maintain a compromise between the desirable sample dilution and the appropriate sensitivity, radial and attenuated radial plasma view were also tested with abovementioned wavelengths. However, only the most sensitive emission line (Fe II 238.204 nm) kept both peaks measurable in broad range. Due to this, other emission lines were not further investigated. Moreover, attenuated radial plasma view allowed to measure samples in proper dilution while radial plasma view required higher dilution factor (the same as the attenuated axial view). According to the above, further analysis was conducted at Fe II 238.204 nm and the attenuated radial plasma view, but the attenuated axial plasma view was occasionally applied (in the case of samples with total iron content less than 2 g kg^−1^).

### 3.2. Analytical Figures of Merit

#### 3.2.1. Detection Limits, Precision and Linear Calibration Range

Peak heights were used for calculations of iron species content due to the almost symmetrical shapes of signals. Detection limits (DLs) in attenuated radial plasma view for Fe(III) were 41 mg kg^−1^ (DF = 50) and 82 mg kg^−1^ (DF = 100), whereas for Fe(II) they were 50 mg kg^−1^ (DF = 50) and 100 mg kg^−1^ (DF = 100). The acceptable correlation coefficient for Fe(III) was R^2^ = 0.998 in the range from DL to 12,500 mg kg^−1^ (DF = 50) and from DL to 25,000 mg kg^−1^ (DF = 100). In the case of Fe(II), R^2^ = 0.981 in the range from DL to 10,000 mg kg^−1^ (DF = 50) and from DL to 20,000 mg kg^−1^ (DF = 100). Precision (as RSD%) was only investigated in the attenuated radial plasma view (as the least sensitive mode) at the highest sample dilution (DF = 100). Measurements were performed with a real sample and acceptable RSD were found (below 10%), i.e., 4.5% for Fe(III) and 5.5% for Fe(II).

In the case of the attenuated axial plasma view (Table 2), DLs were 2.0 mg kg^−1^ and 12 mg kg^−1^ for Fe(III) and Fe(II), respectively, with DF = 50. Linear calibration range for Fe(III) was from DL to 3250 mg kg^−1^ with sufficient correlation coefficient (R^2^ = 0.970) and for Fe(II) from DL to 1500 g kg^−1^ with R^2^ = 0.998. Comparing the DLs of HPLC–ICP OES [35], i.e., 5.41 µg L^−1^ for Fe(III) and 6.31 µg L^−1^ for Fe(II), DLs of HPLC–ICP hrOES were lower in the attenuated axial plasma view, i.e., 0.8 µg L^−1^ and 4.6 µg L^−1^ for Fe(III) and Fe(II), respectively, whereas were higher in the attenuated radial plasma view (16 µg L^−1^ and 20 µg L^−1^).

#### 3.2.2. Standard Addition Method

There are a few certified reference materials (CRMs), which were developed in the 1970s and 1980s, containing certified values of both iron oxides, e.g., rock samples SY-2, SY-3, MRG-1 (CANMET, Hamilton, ON, Canada), SARM 6 and SARM 50 (Mintek, Randburg, South Africa), but these CRMs are not available for sale. There have also been some attempts to create a new certified material––for example, Alcott et al. described a step extraction for iron in rock powder [43]––but this kind of CRM is not appropriate for Fe(II) and Fe(III) speciation analysis because it certifies the total iron amount. Due to the lack of proper CRMs, the standard addition method was performed for the method validation. Therefore, two concentrations (presented as content equivalent in sample, i.e., 5 g kg^−1^ and 10 g kg^−1^) of each iron species were added to one sample of each kind (sediments, soil, and pottery). Adequate volumes (mL) of standards were added during a sample dilution (DF = 100) just before the analysis, and were filled up with water. These analyses were performed in the radial attenuated plasma view. The obtained results are presented in Table 3. All recoveries were in the acceptable range, i.e., 81–120% and 92–104% for Fe(III) and Fe(II), respectively.

### 3.3. Application

An optimized method (HPLC-ICP hrOES) was tested on fifteen samples, on five of each type: sediments (A1–A5), soils (B1–B5), and archaeological pottery (C1–C5). All samples were extracted with 2 mol L^−1^ hydrochloric acid and diluted 50- or 100-fold prior to the analysis. It is worth noting that this procedure does not allow the extraction of Fe(II) and Fe(III) from geological and pottery samples totally. There are some iron species, e.g., those built in silicates, which can only be extracted with HF. Therefore, the total content of iron (which was determined here with ICP hrOES) means that a whole fraction is extracted with HCl solution. All of the results are collected in Table 4.

In most of the samples (9 of 15), there was a dominance of Fe(III), but the content of both species was similar in soils (the ratio between Fe(III) and Fe(II) content ranged from 0.7 to 2.1). Fe(II) was more abundant in one sample (A5). The difference between A5 and the rest of the sediments (A1–A4) could also be indicated by the colour of the samples because only A5 was grey and A1–A4 were brown–red. The content of iron species was summed up and presented as a percentage (%) of the total iron content (which was determined by ICP hrOES), obtaining the acceptable range from 71% to 98%. Iron content (as a total fraction) was also measured and confirmed with ICP (iCRC)MS as a reference technique.

## 4. Conclusions

In this study, a fast and efficient method for speciation analysis of leachable iron in geological and pottery samples was presented. The short cation–exchange column (CG5A) allowed us to separate the Fe(III) and Fe(II) ionic species using the significant low flow rate of the mobile phase (0.5 mL min^−1^). The advantages of the methods used were the shortening of the analysis time, the lower eluent consumption (equivalent to less waste), the lower electricity consumption, and the resulting economic benefits (e.g., lower costs of a single analysis). All of the abovementioned advantages are in line with the principles of green chemistry. The total time of analysis was approx. 5 min, which is very valuable for routine analysis of a big batch of samples. The use of an optical emission spectrometer (ICP hrOES) allowed us to work with high iron concentrations, which is desirable in the case of sediments, soils, or pottery. Moreover, its high resolution optics allowed us to provide more accurate selection of emission lines and to obtain much better DLs (especially in the attenuated axial plasma view) than with a non–high resolution spectrometer. The most sensitive wavelength (Fe II 238.204 nm) was chosen in an attenuated axial as well as in an attenuated radial plasma view. The first mode was better for the very low iron content in samples, and the second mode was the most suitable for more iron-rich samples (e.g., soils or ceramics). The simplicity, availability, and short time of analysis with the short column-based hyphenated system (HPLC–ICP hrOES) suggest that it can be very powerful method for speciation analysis of a big batch of samples, which is desirable due to the need of representativeness of the environmental and archaeological studies.

## Figures and Tables

**Figure 1 molecules-28-04539-f001:**
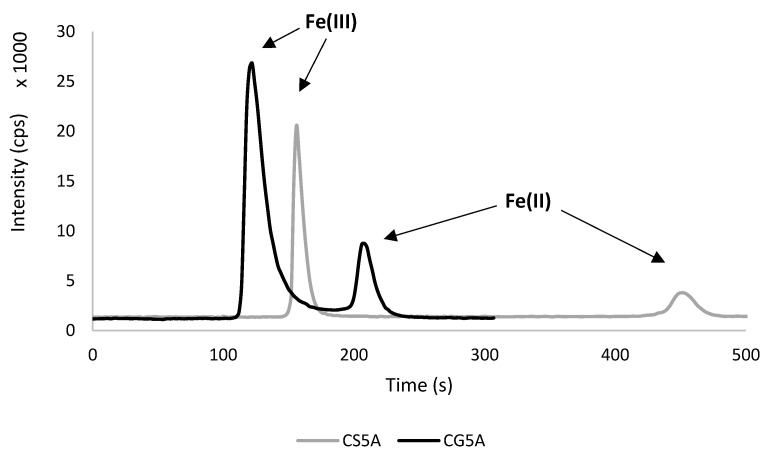
Comparison of CS5A and CG5A column on a real sample (archaeological pottery).

**Figure 2 molecules-28-04539-f002:**
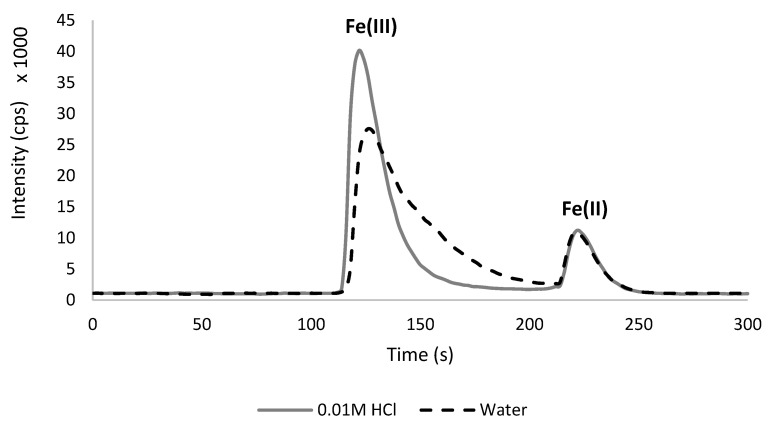
The influence of matrix on the peak intensity (cps) for iron standards—Fe(III) (5 mg L^−1^) and Fe(II) (1 mg L^−1^).

**Table 1 molecules-28-04539-t001:** Experimental conditions for HPLC–ICP hrOES.

HPLC Conditions	
Pump	Varian ProStar 210
Columns	Dionex IonPac CG5A (50 mm × 4.0 mm, i.d.), Dionex IonPac CS5A (250 mm × 4.0 mm, i.d.)
Mobile phases	7.0 mmol L^−1^ PDCA, 66 mmol L^−1^ NaOH, 5.6 mmol L^−1^ Na_2_SO_4_, 74 mmol L^−1^ HCOOH
Mobile phase flow rate (mL min^−1^)	0.5 (CG5A), 2.0 (CS5A)
Injection volume (µL)	200
**ICP hrOES conditions**	
Spectrometer	AnalytikJena PlasmaQuant PQ9000 Elite
RF power (W)	1200
Nebulizer gas flow rate (L min^−1^)	0.7
Auxiliary gas flow rate (L min^−1^)	1.0
Plasma gas flow rate (L min^−1^)	12
Nebulizer/spray chamber type	OneNeb
Torch view	Attenuated radial, attenuated axial
Final analytical wavelength (nm)	Fe II 238.204

**Table 2 molecules-28-04539-t002:** Analytical figures of merit (*n* = 3, peak height).

Species	RT (s)	Emission Line (nm)	Plasma View	Linear Calibration Range (mg kg^−1^)	R^2^	Dilution Factor	DL (mg kg^−1^)	Precision (as RSD) (%)
Fe(III)	123 ± 4	238.204	Attenuated radial	DL–12,500	0.998	50	41	-
				DL–25,000		100	82	4.5
			Attenuated axial	DL–3250	0.970	50	2.0	-
Fe(II)	203 ± 7	238.204	Attenuated radial	DL–10,000	0.981	50	50	-
				DL–20,000		100	100	5.5
			Attenuated axial	DL–1500	0.988	50	12	-

**Table 3 molecules-28-04539-t003:** Recovery at the addition of standard solution (*n* = 3, peak height, DF = 100, radial attenuated).

ICP hrOES	Fe II 238.204 nm			
	Sample Solution (g kg^−1^)	Added (g kg^−1^)	Found (g kg^−1^)	Recovery (%)
A				
Fe(III)	11.2 ± 0.6	5.00	16.5 ± 0.9	97 ± 5
	11.2 ± 0.6	10.0	22.4 ± 1.2	102 ± 5
Fe(II)	0.80 ± 0.05	5.00	6.91 ± 0.44	94 ± 6
	0.80 ± 0.05	10.0	13.2 ± 0.8	95 ± 6
B				
Fe(III)	3.11 ± 0.16	5.00	7.58 ± 0.39	81 ± 4
	3.11 ± 0.16	10.0	15.0 ± 0.8	108 ± 6
Fe(II)	1.37 ± 0.09	5.00	7.94 ± 0.50	101 ± 6
	1.37 ± 0.09	10.0	13.7 ± 0.9	95 ± 6
C				
Fe(III)	13.2 ± 0.7	5.00	19.8 ± 1.0	120 ± 6
	13.2 ± 0.7	10.0	25.6 ± 1.3	113 ± 6
Fe(II)	4.83 ± 0.31	5.00	11.6 ± 0.7	104 ± 7
	4.83 ± 0.31	10.0	16.7 ± 1.1	92 ± 6

A—Sediments, B—Soils, C—Archaeological pottery.

**Table 4 molecules-28-04539-t004:** Applicability of HPLC–ICP hrOES (Fe II 238.204 nm) technique performed on three types of samples (peak height) (g kg^−1^).

Methods				HPLC–ICP hrOES (g kg^−1^)			ICP hrOES (g kg^−1^)	
Sample Matrix	Plasma View	DF	No.	Fe(III)	Fe(II)	Sum	Total Content	Sum as % of Total Content
Sediments	Attenuated axial	50	A1	0.806	0.171	0.977	1.20	81
		50	A2	0.991	0.121	1.11	1.55	72
	Attenuated radial	50	A3	6.86	<DL	6.86	9.64	71
		100	A4	24.4	1.87	26.2	31.3	84
		100	A5	9.93	16.9	26.8	33.0	81
Soil	Attenuated radial	50	B1	2.33	1.13	3.45	3.59	96
		50	B2	4.04	3.18	7.21	8.42	86
		50	B3	2.45	3.43	5.88	6.34	93
		50	B4	0.814	0.823	1.64	2.07	79
		100	B5	2.38	1.34	3.72	3.84	97
Archaeological pottery	Attenuated radial	50	C1	13.9	<DL	13.9	16.6	84
		50	C2	13.1	0.151	13.2	15.9	83
		100	C3	16.0	<DL	16.0	18.7	85
		100	C4	15.5	2.44	17.9	18.2	98
		100	C5	11.6	2.42	14.0	16.4	85

<DL—below method detection limit (including sample preparation and dilution); DF—dilution factor (before an analysis); Total content—a whole iron fraction extracted with HCl solution (2 mol L^−1^).

## Data Availability

Data is contained within the article.

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
