# Peer review of "A Fast and Efficient Procedure of Iron Species Determination Based on HPLC with a Short Column and Detection in High Resolution ICP OES"

_molecules, 2023, doi:10.3390/molecules28114539_

Round 1
Reviewer 1 Report
The optimization of a new hyphenated procedure: HPLC with short cation–exchange column coupled with high resolution inductively coupled plasma optical emission spectrometry for for iron ionic speciation, iss presented by authors of this paper. The applicability on three types of real samples :sediments, soils and archaeological pottery, was presented. Introduction is clear and present almost all method used in speciation of iron ions. Materials and methods are well presented. Results and discussion are divided in sub-chapters in which optimization process, analytical figures of merit, and application results are presented. Authors concluded that the simplicity, availability and short time of analysis can be very powerful technique for speciation analysis of different samples.
Author Response
Reviewer’s Comments: The optimization of a new hyphenated procedure: HPLC with short cation–exchange column coupled with high resolution inductively coupled plasma optical emission spectrometry for for iron ionic speciation, iss presented by authors of this paper. The applicability on three types of real samples :sediments, soils and archaeological pottery, was presented. Introduction is clear and present almost all method used in speciation of iron ions. Materials and methods are well presented. Results and discussion are divided in sub-chapters in which optimization process, analytical figures of merit, and application results are presented. Authors concluded that the simplicity, availability and short time of analysis can be very powerful technique for speciation analysis of different samples.
Authors’ Response: We sincerely thank the Reviewer for the positive acceptance of our manuscript.
Reviewer 2 Report
The optimization and application of a HPLC-ICP-OES procedure for iron ionic speciation is presented in this manuscript. Although the manuscript is well-written and scientific sound, the manuscript's contribution to the field appears to be somewhat limited in terms of novelty.
The manuscript is essentially based on a very simple evolution of an existing HPLC-ICP-OES strategy previously developed by the same authors (DOI: 10.1016/j.talanta.2021.122403). The simple shortening of the cation-exchange column clearly provides advantages in terms of analysis time and solvent saving: this effect is obvious and well-known within the scientific community. Thus, this presented advance does not in itself represent such a significant discovery that it merits publication of the submitted manuscript.
Moreover, the lack of novelty is also evident under the chemistry of separation: the employment of PDCA-based eluent coupled with cation-exchange stationary phases is already known in the scientific literature by at least 30 years! (see DOI: 10.1016/S0021-9673(01)93022-5, 10.1016/S0021-9673(99)00426-4 plus references mentioned in the presented manuscript). Concerning the coupling of the HPLC system with ICP-OES rather than using an UV detector (after post column reaction), this evolution is already demonstrated in the literature, specifically also by the authors of this article.
In view of all these aspects, I cannot recommend the publication of this manuscript in Molecules due to the absence of significant progress from the analytical chemistry’s point of view.
Author Response
Reviewer’s Comment #1: The optimization and application of a HPLC-ICP-OES procedure for iron ionic speciation is presented in this manuscript. Although the manuscript is well-written and scientific sound, the manuscript's contribution to the field appears to be somewhat limited in terms of novelty.
Authors’ Response #1: We sincerely thank the Reviewer for the positive reception of our manuscript. We hope that our comments and the revised text will meet the reviewer's expectations in terms of the scientific novelty. The following statements were added in the text:
Lines 93-96: In this study, the first combination of short cation–exchange HPLC column (50 mm x 4 mm) with detection in high resolution ICP OES (ICP hrOES) was presented. The novel system was properly optimized and then compared with previously presented HPLC–ICP OES [35].
Lines 100-102: The introduction of short column HPLC coupled to ICP hrOES was reported as a missing tool in hyphenated techniques for determination of inorganic iron species, Fe(II) and Fe(III).
Lines 157-161: Although conditions and parameters of this technique (HPLC–ICP hrOES) were based on previously optimised system (HPLC–ICP OES) [35], it was constructed from the basis and all crucial parts, i.e. a HPLC pump, a HPLC column, and a detector was different. Therefore, significant improvements have been proposed both in terms of the chromatographic run and the spectrometric detection.
Lines 167-168: According to the above, high resolution ICP OES was a much more resistant detector to signal interferences than this conventional one.
Lines 184-185: Nevertheless, this modification of mobile phase was recognized as a significant improvement of commercial available eluent and it was used for further research.
Lines 225-227: This influence of matrix in the preparation of calibration standard was probably observed for the first time in the literature.
Lines 234-236: Worth notice that three modes of plasma view were optimized for the first time in iron speciation analysis based on hyphenated techniques.
Reviewer’s Comment #2: The manuscript is essentially based on a very simple evolution of an existing HPLC-ICP-OES strategy previously developed by the same authors (DOI: 10.1016/j.talanta.2021.122403). The simple shortening of the cation-exchange column clearly provides advantages in terms of analysis time and solvent saving: this effect is obvious and well-known within the scientific community. Thus, this presented advance does not in itself represent such a significant discovery that it merits publication of the submitted manuscript.
Authors’ Response #2: With all due respect, we cannot agree with the Reviewer. Changing the length of the analytical column is only seemingly trivial. Changed chromatographic separation parameters require changes in both the concentration (pH, temperature) of the eluent and its flow. This makes it difficult to interface with the ICP OES sample introduction system. Rapid changes in the analytical signal may require a higher resolution of spectrometer. Therefore, the constructed chromatographic system becomes a completely new challenge when creating a methodology.
The following system (HPLC-ICP hrOES) was constructed from the basis developing this new analytical tool both from the chromatographic run (from long to short HPLC column) and spectrometric detection (from conventional to high resolution ICP OES). Therefore, all crucial parts (i.e., a HPLC pump, a HPLC column, and a high resolution ICP OES) are different in comparison with previously published system (Proch & Niedzielski 2021). Moreover, we optimized parameters (like plasma view, a modification of commercial eluent), which stayed fixed in previous work. We also reported the matrix effect in calibration standard preparation. In addition, scientific community has not published any similar systems yet. Therefore we are surprised for whom advantages in terms of analysis time and solvent saving are well-known if a long column-based HPLC systems are still published papers and dominant in the literature. Moreover, this hyphenated system was operating with better parameters (in terms of analysis time and solvent saving) than all short column-based HPLC systems (Wolle et al. 2014, Fernserner et al. 2014), which had been reported in the text.
Reviewer’s Comment #3: Moreover, the lack of novelty is also evident under the chemistry of separation: the employment of PDCA-based eluent coupled with cation-exchange stationary phases is already known in the scientific literature by at least 30 years! (see DOI: 10.1016/S0021-9673(01)93022-5, 10.1016/S0021-9673(99)00426-4 plus references mentioned in the presented manuscript). Concerning the coupling of the HPLC system with ICP-OES rather than using an UV detector (after post column reaction), this evolution is already demonstrated in the literature, specifically also by the authors of this article.
Authors’ Response #3: We would like to thank the Reviewer for pointing out that an important work (Ruth & Shaw 1991 [28]) has been omitted. This has been corrected.
Lines 68-70: Other reagents are variable and depend on chosen methods as well as analysed materials, e.g., lithium hydroxide, sodium sulfite, sodium chloride [28], acetic acid, sodium acetate, ascorbic acid [24], potassium chloride, formic acid, potassium hydroxide [27] or potassium sulfate, potassium hydroxide and formic acid, which is a commercially available mobile phase dedicated for CS5A column [23,25,29–32].
The second article (Cardellicchio et al. 1999, [23]) has been already cited as well as the first use of HPLC with PDCA-based eluent by Cardellicchio et al. (1997) [40], DOI: 10.1016/S0021-9673(97)00086-1.
However we cannot agree with the further part of this comment. This paper does not report the evolution between UV detection and ICP OES and the novelty is a replacing of conventional ICP OES with high resolution ICP OES as a HPLC detector. Changing the detector to the high-resolution ICP hrOES, just like changing the length of the column, is only seemingly trivial. According to Scopus database, keywords "high resolution ICP OES" containing 15 papers only with no chromatographic systems and one paper focusing on a micro solid phase extractions for Cd and Pb speciation analysis (DOI: 10.14233/ajchem.2021.23224). In comparison to keywords “high resolution ICP MS” or “high resolution ICP MS” AND “chromatography” there are 183 or 29 papers respectively. All above mentioned bibliometric data confirms that a hyphenated system with detection in high resolution ICP OES is a scientific novelty.
Reviewer’s Comment #4: In view of all these aspects, I cannot recommend the publication of this manuscript in Molecules due to the absence of significant progress from the analytical chemistry’s point of view.
Authors’ Response #4: Thank you for your thorough and critical evaluation of the text. The article has been corrected as indicated by the reviewer. We hope that the revised text will meet the criteria of being accepted for publication.
Round 2
Reviewer 2 Report
I would like to express again my appreciation for the well-structured scientific article and evidence presented that is methodically obtained. I believe that the novelty of the work is still limited. However, the authors' comments and responses were satisfactory providing a critical and interesting discussion that facilitated the acceptance of this article for publication. Therefore, since any doubts have been addressed with clarifications where necessary, the manuscript in its current form can be accepted for publication.